Mass marking of juvenile Schizothorax wangchiachii (Fang) with alizarin red S and evaluation of stock enhancement in the Jinping area of the Yalong River

Yang Kun 1
Li Shu 1
Liu Xiaoshuai 2
Gan Weixiong 2
Deng Longjun 2
Tang Yezhong 3
Song Zhaobin zbsong@scu.edu.cn 1 4
1 College of Life Sciences, Sichuan University, Sichuan Key Laboratory of Conservation Biology on Endangered Wildlife , Chengdu , Sichuan Province , People’s Republic of China
2 Yalong River Hydropower Development Company, Ltd. , Chengdu , Sichuan Province , People’s Republic of China
3 Chinese Academy of Sciences, Chengdu Institute of Biology , Chengdu , Sichuan Province , People’s Republic of China
4 College of Life Sciences, Sichuan University, Key Laboratory of Bio-Resources and Eco-Environment of Ministry of Education , Chengdu , Sichuan Province , People’s Republic of China
Kramer Donald
Electronic publication date: 2017 Dec 6
Publication date: 2017
Volume: 5
Electronic Location ID: e4142
Received 2017 Oct 6; Accepted 2017 Nov 15
Copyright: ©2017 Yang et al.
Copyright year: 2017
Copyright holder: Yang et al.
License: This is an open access article distributed under the terms of the Creative Commons Attribution License, which permits unrestricted use, distribution, reproduction and adaptation in any medium and for any purpose provided that it is properly attributed. For attribution, the original author(s), title, publication source (PeerJ) and either DOI or URL of the article must be cited.
License URL: https://creativecommons.org/licenses/by/4.0/

Keywords: Stock enhancement, Otolith mass marking, Recapture survey

Funding: Yalong River Hydropower Development Company, Ltd. 12H0856 Program for New Century Excellent Talents in University NCET-11-0347 This study was supported by the Yalong River Hydropower Development Company, Ltd. (grant number 12H0856) and Program for New Century Excellent Talents in University (grant number NCET-11-0347). The funders had no role in study design, data collection and analysis, decision to publish, or preparation of the manuscript.

==============================
Schizothorax wangchiachii is a key fish species in the stock enhancement program of the Yalong River hydropower project, China. Alizarin red S (ARS) was used to mark large numbers of juvenile S. wangchiachii in the Jinping Hatchery and later used to evaluate stock enhancement in the Jinping area of the Yalong River. In a small-scale pilot study, 7,000 juveniles of the 2014 cohort were successfully marked by immersion in ARS solution, and no mortality was recorded during the marking process. The ARS mark in the fish otoliths remained visible 20 months later. In the large-scale marking study, approximately 600,000 juveniles of the 2015 cohort were successfully marked. Mortalities of both marked and unmarked juveniles were very low and did not differ significantly. Total length, wet mass and condition factor did not differ significantly between unmarked and marked individuals after three months. On 24 July 2015, about 840,000 Jinping Hatchery-produced young S. wangchiachii, including 400,000 marked individuals, were released at two sites in the Jinping area. Recapture surveys showed that (1) marked and unmarked S. wangchiachii did not differ significantly in total length, wet mass and condition factor; (2) stocked individuals became an important part of recruitment of the 2015 cohort; (3) instantaneous growth rate of marked individuals tended to slightly increase; and (4) most stocked individuals were distributed along a 10–15 km stretch near the release sites. These results suggest that the ARS method is a cost-efficient way to mass mark juvenile S. wangchiachii and that releasing juveniles is an effective means of stock recruitment.

Introduction

Habitat degradation and overexploitation contribute to the decline of fisheries around the world. In order to improve fish stocks, stock enhancement by releasing hatchery-produced fish into wild habitats has been widely implemented (Brown & Day, 2002; Taylor et al., 2017; Yang et al., 2013). The effectiveness of stock enhancement can be assessed by mark-release-recapture studies (Blaxter, 2000), which require effective tagging or marking methods. Various marking techniques, such as otolith marking (Volk, Schroder & Grimm, 1999), coded wire tags (Bernard, Marshall & Clark, 1998) and passive integrated transponders (Navarro et al., 2006), have been developed to monitor released fish. Among these techniques, otolith marking is a feasible method that allows long-term identification of small fish (Caraguel et al., 2015; Crook et al., 2009). Otolith marks can be achieved by fluorescent marking (Yang et al., 2016), thermal marking (Volk, Schroder & Grimm, 1999) and isotopic marking (Woodcock et al., 2011). The most popular marking protocol is to use fluorochromes, which can form chelate complexes with calcium ions that are built into skeletal and otolith structures (Poczyczyński et al., 2011). Fluorescent marks on the otolith are visible under a specific inducing laser because the calcium-fluorochrome complexes emit fluorescent light (Bashey, 2004; Taylor, Fielder & Suthers, 2005; Yang et al., 2016). Fluorochromes commonly used for otolith marking are alizarin red S (ARS), alizarin complexone, oxytetracycline hydrochloride and calcein. Compared with other fluorochromes, ARS offers better mark quality and lower cost and thus is viewed as a promising dye for mass marking fish at early life stages (Taylor, Fielder & Suthers, 2005; Yang et al., 2016).

In China, releasing hatchery-reared fish to enhance or restore fish stock abundance and fishery catches has been widely implemented for more than fifty years (Yang et al., 2013). Chinese carps and several other commonly cultured species that do not breed effectively in still waters were selected for early artificial rearing-releasing programs (Wu & Zhong, 1964; Liu, 1965). In recent decades, technical developments and advances in hatchery production have made it possible to breed considerable numbers of endemic and rare fish annually, including Chinese sturgeon Acipenser sinensis (Chang & Cao, 1999) and Chinese sucker Myxocyprinus asiaticus (Zhou et al., 1999). However, fish stock enhancement programs in China, particularly for freshwater fish species, have focused mainly on artificial propagation techniques and stocking scale, with little attention paid to monitoring and evaluating the success of fish after release (Cheng & Jiang, 2010; Yang et al., 2013; Zhang, Li & Shu, 2003). Some marking technologies have been tested in different fish species in recent years, but rarely has large-scale marking and recapturing been used in stock enhancement (Zhang, Li & Shu, 2003; Yang et al., 2013). Therefore, it is necessary to carry out post-release evaluation based on mass marking and recapturing.

Figure 1 Map of the Jinping area of the Yalong River showing the locations of Jinping Hatchery and the sites where the stocked S. wangchiachii were released and recaptured.

Fish were released at sites 2 and 3 in July 2015, recaptures were conducted at sites 1–7 from October 2015 to April 2016. Base maps are available from NFGIS (National Fundamental Geographic Information System, http://nfgis.nsdi.gov.cn/).

Schizothorax wangchiachii (Fang, 1936), which belongs to the subfamily Schizothoracinae of the family Cyprinidae, is distributed mainly in the upper Yangtze River and its tributary, the Yalong River (Yue, 2000). This species is adapted to torrential mountain rivers in the southeastern Qinghai-Tibetan Plateau (Yue, 2000). Before the 1990s, S. wangchiachii was caught abundantly in many parts of its distribution range. However, its recruitment has declined dramatically since the mid-1990s, likely due to habitat degradation, overfishing and hydropower development (Deng, Yu & Li, 2000; Duan, Deng & Ye, 1995; Jiang et al., 2007). To improve the health of the S. wangchiachii population in the Yalong River, conservation plans, such as building fish hatcheries, have been initiated (Wang, Wu & Deng, 2011).

Jinping Hatchery (28°18′39.09″N, 101°38′50.10″E; Fig. 1) is the first and most important fish hatchery located in the lower Yalong River. The hatchery is used to domesticate and propagate S. wangchiachii and many other fish species that are threatened by hydropower development in the Yalong River. Since 2011, annual release of S. wangchiachii (total length 40–80 mm) from Jinping Hatchery has been carried out in the Jinping area of the Yalong River (Deng, Wang & Gan, 2016). The objectives of this study were to assess the feasibility of mass marking S. wangchiachii using ARS and subsequently to evaluate the effectiveness of stock enhancement by recapturing marked individuals of the 2015 cohort after release.

Materials and Methods

Experimental fish and fluorochrome

Juveniles of S. wangchiachii used for marking and stocking in this study were produced at Jinping Hatchery using an artificial propagation technique. The breeding stocks were native spawners caught from the wild in the Jinping area of the Yalong River in 2011 and 2012. Juveniles were reared in numerous cylindrical tanks in the juvenile rearing room. These tanks are made of fiberglass, have a diameter of 2 m and a height of 1 m, and each has a temperature-controlled water supply (water temperature, 15.5–17.5 °C; dissolved oxygen concentration, 7.0–7.4 mg L−1; pH, 7.1–7.4) from the recirculating aquaculture system.

The fluorochrome ARS (C14H7NaO7S) used for marking was analytically purified powder. During immersion marking, ARS was dissolved directly in the rearing water according to the experimental design. To optimize marking quality and minimize juvenile mortality, several preliminary experiments were conducted in 2013, and results showed that immersing juvenile S. wangchiachii in water containing ARS doses ≤100 mg L−1 for 24 h resulted in the lowest death rate while producing a mark that could be seen clearly in the otolith.

Small-scale marking pilot study

A small-scale marking pilot study was performed in the rearing tank. Approximately 7,000 40-day-old juvenile S. wangchiachii of the 2014 cohort (total length 23.52 ± 1.50 mm, mean ± S.D., n = 20) reared in a tank were selected for immersion marking on 5 May 2014. These juveniles were starved for 24 h prior to the treatment. The inner wall of the tank was carefully cleaned, and the rearing water was completely replaced with about 500 L of clean water. Thirty-five grams of ARS were pre-dissolved in about 10 L of water, which was immediately added to the tank. The juveniles were immersed in the ARS solution (70 mg L−1) for 24 h. During immersion, the solution was aerated continuously and the fish were not fed. After immersion was completed, the ARS solution was discharged into a sewage pool, and at the same time clean water was pumped into the tank to thoroughly rinse out the remnant dye. One day after immersion, dead individuals were counted and removed from the tank. To check the visibility and persistence of the marks, 10 marked fish from the tank were haphazardly sampled and sacrificed with an overdose of MS-222 (100 mg L−1) on 29 May 2014, 28 December 2014, 4 May 2015 and 24 January 2016. Sampled fish were kept in 100% ethanol until otolith examination.

Large-scale marking application

From late April to early May 2015, five batches of juvenile S. wangchiachii (total length 20.85 ± 1.41 mm, mean ± S.D., n = 140) of the 2015 cohort were marked at Jinping Hatchery using the ARS immersion protocol described above. In total, an estimated 600,000 fish were marked. To ensure the safety of the very small juveniles during immersion marking, the concentrations of ARS solution were controlled within a range of 30–50 mg L−1. One day after immersion was completed, dead fish in each tank were counted and recorded. The mortalities of three batches of unmarked fish in the rearing room also were recorded. Five days after marking, both marked and unmarked fish were transferred to four outdoor fishponds in the hatchery. The fish were fed to satiation three times a day with a commercial artificial compound diet.

To assess effects of the marking process on growth, 100 marked and unmarked S. wangchiachii at similar daily age were haphazardly taken from two fishponds on 24 July 2015. These fish were starved for 24 h prior to further treatment. Afterwards, they were anaesthetized with MS-222 at a concentration of 100 mg L−1. Their total length and standard length were measured to the nearest 0.01 mm with a digital caliper, and their wet mass was weighed to the nearest 0.0001 g with a precision electronic balance. To assess the mark effectiveness, 400 marked fish also were selected haphazardly from those in the four fishponds. They were anaesthetized with MS-222 at a concentration of 100 mg L−1 and then stored in 100% ethanol until otolith examination.

Release and recapture

On 24 July 2015, 840,000 young S. wangchiachii of the 2015 cohort, of which 400,000 individuals were had been marked by ARS, were released at sites 2 and 3 in the Jinping area of the Yalong River (Fig. 1). Site 2 (28°18′47.80″N, 101°38′51.19″E) is located in the wide and deep part of the river (>5 m maximum depth). Although it was not a suitable habitat for S. wangchiachii, site 2 was used because of stairs that provided access to the river. Site 3 (28°19′41.32″N, 101°38′52.18″E) was near a sand quarry, and the substrate was covered with gravel and small stones: this was an appropriate habitat for young fish. The distance between the two release sites was about 2 km. The fish were first captured from each fishpond with a nylon trawl, and 150 marked individuals were haphazardly selected for measuring and weighing. Afterwards, fish were transferred to the release sites by a pickup truck and released into the river using buckets. For comparison of growth, about 10,000 unmarked S. wangchiachii were raised as a control group in a fishpond in Jinping Hatchery. They were fed twice daily with commercial feed at a ratio of about 3% of body weight per day.

Before the recapture surveys were conducted, appropriate recapture sites along the Jinping area of the Yalong River were chosen. Criteria included ease of fishing, suitable habitats for young fish and the distances from the release sites. Seven sites, including the two release sites, along a 60 km stretch of the Yalong River in the Jinping area were selected for the recapture surveys (Fig. 1). Site 1 (28°17′46.38″N, 101°38′39.19″E) was a shallow riffle area (<0.5 m in depth) with a substrate of gravel and small stones located about 1.5 km upstream of site 2 and 7 km downstream of Jinping Dam II. Site 1 was an important nursery ground for Schizothorax fish at early life stages. Site 4 (28°20′50.68″N, 101°39′18.04″E), site 5 (28°24′10.28″N, 101°43′24.86″E), site 6 (28°27′41.38″N, 101°44′49.65″E) and site 7 (28°36′58.04″N, 101°55′56.04″E) were located about 3, 15, 20 and 50 km downstream of site 3, respectively, and had a similar substrate of small stones and occasional boulders.

Recapture surveys were carried out at three-month intervals (in October 2015, January 2016 and April 2016) using a similar method and sampling effort each time. At each site, three 9-m long fishing pots with 6-mm mesh were used for recapturing fish for four successive days, and catches in each pot were removed once a day. The fishing pot used in these surveys was a trap-type stationary fishing device that was especially suitable for catching small fish with total length <20 cm. Electrofishing permitted by the Sichuan Municipal Bureau of Aquatic Products was performed only one time at each site, and it involved using a 30-cm-diameter anode and a 6-mm mesh landing net to sample for 40 min along the river. Specimens of S. wangchiachii assumed to be from the 2015 cohort based on personal experience of age-total length were sacrificed with an overdose of MS-222 and measured and weighed, whereas other fish were released. Meanwhile, 50 individuals sampled from the hatchery control group also were measured. All sampled fish were stored in 100% ethanol for further processing.

Otolith removal and examination

Three pairs of otoliths were removed from all fish sampled, and the left three otoliths were mounted on glass slides using neutral balsam. To check the ARS mark and to read age, otoliths were observed under an Olympus BX40 fluorescence microscope fitted with a Q-Imaging MicroPublisher 5.0 RTV digital camera using the green laser and normal transmitted light (Yang et al., 2016).

Data analysis

In this study, instantaneous growth rate of mean total length (Gl) was calculated following Ricker (1975) as: Gl=lnl2−lnl1∕3

where l1 is the mean total length in millimeters of S. wangchiachii of at a given time point, l2 is the corresponding mean total length of the same batch three months later, and 3 is the sampling interval of three months.

Instantaneous growth rate of mean wet mass (Gw) was calculated as: Gw=lnw2−lnw1∕3

where w1 is the mean wet mass in grams of S. wangchiachii at a given time, w2 is the corresponding mean wet mass of the same batch three months later, and 3 is the sampling interval of three months.

The percent (Pi) of marked individuals out of all marked S. wangchiachii for each recapture survey was calculated as: Pi=ni∕Nt

where ni is the number of marked fish at site i of the given recapture date and Nt is the total number of marked fish of the given recapture.

The condition factor of S. wangchiachii was calculated following Fulton (1904) as: condition factor=106×w∕l3

where w is the wet mass in grams of S. wangchiachii at a given time point and l is the total length in millimeters of S. wangchiachii at the same time.

ANOVA tests for total length, wet mass and condition factor between marked and unmarked S. wangchiachii were performed with SPSS 19.0 software. Bonferroni Tests were used for post hoc tests when the variances were equal, and alternatively, Games-Howell Tests were used when the variances were unequal. The significance level was set as P < 0.05.

Results

Small-scale marking pilot study

One day after immersion was completed, no marked fish had died. Twenty-three days after marking, all sampled individuals showed a visible red-orange mark in their otoliths (Fig. 2A). Without polishing, visible marks also were easily identified in the otoliths of marked individuals sampled on 28 December 2014 (Fig. 2B), 4 May 2015 (Fig. 2C) and 24 January 2016 (Fig. 2D). More than one year after immersion, the ARS mark still remained visible, and there was no evidence that the mark was significantly decaying.

Figure 2 Photographs of lapillus otoliths of ARS marked S. wangchiachii of the 2014 cohort sampled on 29 May 2014 (A), 28 December 2014 (B), 4 May 2015 (C) and 24 January 2016 (D).

These fish were marked by immersing in 70 mg L−1 ARS for 24 h on 5 May 2014. Photographs were taken under green laser and ×40 magnification. White arrows show the ARS marks.

Large-scale marking application

In the large-scale immersion marking of juvenile S. wangchiachii, mortalities in each batch were very low (≤0.50%), and no significant difference in the mortality between marked and unmarked batches was detected (Table 1). Three months after being reared in outdoor fishponds, samples of marked and unmarked individuals showed no significant difference in total length, wet mass and condition factor (Table 1). A visible red-orange mark in the otoliths could be identified under green laser in all marked fish sampled.

Figure 3 Photographs of ARS marked (A) and unmarked (B) lapillus otolith of S. wangchiachii of the 2015 cohort recaptured in the Jinping area of the Yalong River.

Photographs were taken under green laser and ×40 magnification. The white arrow shows the ARS marks.

Table 1 The mortalities of juvenile S. wangchiachii unmarked and marked by immersion in late April to early May 2015, and total length, wet mass and condition factor of these fish three months later.

Mortality (mean and S.D.) of three batches of unmarked juvenile S. wangchiachii and five batches marked by immersion in 30–50 mg L−1 ARS solution for 24 h in late April to early May 2015, was very low and did not differ significantly (independent t-test, P = 0.836) between treatments. After being reared for three months in outdoor fishponds, total length (P = 0.936), wet mass (P = 0.629) and condition factor (P = 0.244) (mean and S.D.) of marked and unmarked S. wangchiachii did not differ significantly (independent t-test).

Samples	N	Acute mortality (%)	Total length (mm)	Wet mass (g)	Condition factor (%)	
Marked batches	5	0.16(0.20)				
Unmarked batches	3	0.19(0.20)				
Marked individuals	100		49.17(9.76)	1.1460(0.6563)	1.715(0.175)	
Unmarked individuals	100		49.07(7.16)	1.1052(0.5288)	1.748(0.212)	
Notes.

N number of samples

Table 2 Mean total length, wet mass and condition factor of marked and unmarked S. wangchiachii of the 2015 cohort.

These data were obtained at the time of initial release and from recapture surveys carried out in October 2015 and January and April 2016 at seven sites in the Jinping area of the Yalong River. The numbers in parentheses are the standard deviations.

Source	Marked fish	Unmarked fish	
	N.	Total length (mm)	Wet mass (g)	Condition factor	N	Total length (mm)	Wet mass (g)	Condition factor	
Jul. 2015	
Initial release fish	150	44.94(6.67)	0.8225(0.3332)	1.762(0.373)					
Hatchery control fish					50	41.54(4.91)	0.6740(0.2786)	1.893(0.762)	
Oct. 2015	
Hatchery control fish					50	63.79(11.26)	2.9886(1.8327)	1.995(0.431)	
Recaptured fish									
Site 1	45	52.21(7.34)	1.0341(0.4257)	1.430(0.170)	69	54.61(6.72)	1.1551(0.4236)	1.447(0.193)	
Site 2	31	53.02(5.61)	1.0889(0.4352)	1.472(0.115)	51	57.27(9.87)	1.5174(1.0875)	1.518(0.133)	
Site 3	48	54.03(7.11)	1.1733(0.5013)	1.405(0.187)	118	55.88(6.96)	1.2401(0.4663)	1.349(0.151)	
Site 4	6	59.30(6.51)	1.5886(0.6685)	1.400(0.075)	20	58.85(6.36)	1.4423(0.4303)	1.392(0.179)	
Site 5	27	51.04(6.32)	0.9882(0.3255)	1.482(0.184)	68	51.54(5.61)	1.0309(0.3654)	1.481(0.195)	
Site 6	6	59.88(13.19)	1.8959(1.3197)	1.511(0.158)	9	52.46(5.83)	1.1436(0.3253)	1.570(0.153)	
Site 7	1	55.36	1.4478	1.835	2	53.58(3.76)	1.3499(0.3667)	1.618(0.047)	
Total	164	53.26(7.23)	1.1320(0.5263)	1.443(0.169)	337	55.03(7.38)	1.2325(0.5939)	1.431(0.182)	
Jan. 2016	
Hatchery control fish					50	88.20 (7.77)	6.4593(1.5860)	1.770(0.261)	
Recaptured fish									
Site 1	–				–				
Site 2	0				2	67.13(17.12)	2.2926(1.7039)	1.604(0.265)	
Site 3	17	53.57(8.82)	1.3193(0.7693)	1.648(0.186)	36	56.87(7.22)	1.5691(0.7476)	1.676(0.157)	
Site 4	21	67.51(15.57)	2.9274(2.1312)	1.622(0.212)	77	67.70(14.15)	2.9843(2.1267)	1.705(0.150)	
Site 5	9	70.02(8.99)	3.1357(1.2962)	1.867(0.145)	17	74.55(9.04)	4.1380(1.6331)	1.989(0.245)	
Site 6	7	75.37(9.24)	3.5619(1.4448)	1.749(0.118)	18	69.63(8.20)	2.9381(0.9454)	1.823(0.199)	
Site 7	4	76.28(3.93)	4.1186(0.7443)	1.996(0.209)	12	72.64(9.62)	3.4644(1.6936)	1.814(0.162)	
Total	58	65.37(13.98)	2.6471(1.7583)	1.709(0.214)	162	66.59(12.70)	2.8128(1.8472)	1.748(0.194)	
Apr. 2016	
Recaptured fish									
Site 1	–				–				
Site 2	0				4	65.62 (12.48)	2.7153(1.7478)	1.688(0.104)	
Site 3	11	75.28(10.18)	3.8435(1.4643)	1.553(0.094)	20	77.58(14.38)	4.1291(3.2624)	1.450(0.173)	
Site 4	22	85.22(17.52)	6.3145(4.1856)	1.653(0.125)	45	79.16(9.76)	4.5388(2.0997)	1.628(0.116)	
Site 5	1	96.71	9.1327	1.934	11	99.50(15.42)	10.1536(4.5667)	1.805(0.143)	
Site 6	4	104.02(5.54)	10.2389(1.4235)	1.764(0.126)	11	94.99(13.61)	7.7649(3.4934)	1.627(0.164)	
Site 7	2	93.32(13.11)	7.5525(3.2873)	1.671(0.076)	0				
Total	40	85.06(16.42)	6.1598(3.7384)	1.644 ±0.134)	91	82.59(14.84)	5.4373(3.5597)	1.613(0.171)	
Notes.

– no fish were captured because we were unable to get to the site

N number of fish

Recapture and evaluation

Otolith checking confirmed that a total of 852 S. wangchiachii of the 2015 cohort were caught during the three recapture surveys. Of these fish, 262 individuals had a clear ARS mark in their otoliths, and 590 individuals had no mark (Fig. 3; Table 2). The percent of marked individuals in each recapture survey were 32.73% in October 2015, 26.36% in January 2016 and 30.53% in April 2016.

In the October 2015 recapture effort, total length, wet mass and condition factor of both marked and unmarked S. wangchiachii were significantly lower than those of the hatchery control group (Table 2; one-way-ANOVA with post hoc Games-Howell Tests, P < 0.001). The condition factor of marked individuals when they were recaptured was also significantly lower than when they were released (Table 2; independent t-test, P < 0.001). Between marked and unmarked individuals, there was no significant difference in wet mass (P = 0.134) and condition factor (P = 0.735), whereas a slight difference in total length was detected (P = 0.030). In addition, the total length, wet mass and condition factor differed significantly among samples caught from different recapture sites (Table 3; two-way ANOVA tests without the data for site 7, P < 0.001), but a significant difference was not observed between marked and unmarked individuals (P > 0.05). Recapture site and ARS mark had interaction effects on wet mass (P = 0.006).

Table 3 The results of Two-way ANOVAs on the effects of recapture site (R) and ARS mark (A) and their interaction (R × A) on total length, wet mass and condition factor of S. wangchiachii of the 2015 cohort.

Original data were obtained from recapture surveys carried out in October 2015 and January and April 2016 at seven sites in the Jinping area of the Yalong River.

Source	Dependent variable	SS	df	F	P	
Oct. 2015						
Recapture site	Total length	1,362.827	5	272.565	<0.001	
	Wet mass	8.040	5	1.608	<0.001	
	Condition factor	1.091	5	0.218	<0.001	
ARS mark	Total length	1.948	1	1.948	0.844	
	Wet mass	0.086	1	0.086	0.596	
	Condition factor	0.005	1	0.005	0.683	
R × A	Total length	487.183	5	97.437	0.088	
	Wet mass	5.087	5	1.017	0.006	
	Condition factor	0.168	5	0.034	0.316	
Error	Total length	24,574.918	486			
	Wet mass	149.255	486			
	Condition factor	13.776	486			
Jan. 2016						
Recapture site	Total length	8,178.913	4	15.296	<0.001	
	Wet mass	118.057	4	10.572	<0.001	
	Condition factor	1.860	4	15.275	<0.001	
ARS Mark	Total length	2.128	1	0.016	0.900	
	Wet mass	0.001	1	<0.001	0.984	
	Condition factor	0.019	1	0.612	0.435	
R × A	Total length	433.993	4	0.812	0.519	
	Wet mass	9.423	4	0.844	0.499	
	Condition factor	0.218	4	1.792	0.132	
Error	Total length	27,805.420	208			
	Wet mass	580.665	208			
	Condition factor	6.331	208			
Apr. 2016						
Recapture site	Total length	4,442.714	2	13.549	<0.001	
	Wet mass	208.859	2	12.230	<0.001	
	Condition factor	0.476	2	13.454	<0.001	
ARS Mark	Total length	297.492	1	1.815	0.181	
	Wet mass	28.595	1	3.349	0.070	
	Condition factor	0.126	1	7.134	0.009	
R × A	Total length	418.750	2	1.277	0.283	
	Wet mass	25.205	2	1.476	0.233	
	Condition factor	0.049	2	1.388	0.254	
Error	Total length	17,542.681	107			
	Wet mass	913.667	107			
	Condition factor	1.895	107			

In the January 2016 recapture effort, the total length and wet mass of both marked and unmarked S. wangchiachii were still significantly lower than those of the hatchery control group (Table 2; one-way-ANOVA with post hoc Games-Howell Tests, P < 0.001), but the difference in condition factor was not significant (P = 0.304). Between marked and unmarked individuals, no significant differences in total length (P = 0.828) and wet mass (P = 0.816) were detected. The total length, wet mass and condition factor of S. wangchiachii differed significantly among samples caught from different recapture sites (Table 3; two-way ANOVA tests without the data for sites 1 and 7, P < 0.001), but ARS mark and the interaction between it and recapture site did not have significant effects on the three indexes (P > 0.05).

In the April 2016 recapture effort, the hatchery control group was not sampled because of fishpond cleaning. There was no significant difference in total length, wet mass or condition factor between unmarked and marked S. wangchiachii (Table 2; independent t-test, P > 0.05). The total length, wet mass and condition factor of S. wangchiachii differed significantly among samples caught from different recapture sites (Table 3; two-way ANOVA tests, including only data for sites 3, 4 and 6, P < 0.001). A significant difference in condition factor (Table 3; P = 0.009) was observed between marked and unmarked individuals, but a significant difference was not observed for total length (P = 0.181) or wet mass (P = 0.07). The interaction effects of recapture site and ARS mark were not significant (P > 0.05).

The Gl and Gw of marked individuals tended to slowly increase after release. In the first trimester after release, the Gl and Gw values of marked individuals were 0.0566 and 0.1065, respectively, which were lower than those of the hatchery control group (0.1430 and 0.4964, respectively). In the second trimester, both Gl (0.0683) and Gw (0.2832) of marked individuals had increased slightly. At that time, the Gw of marked individuals was slightly higher than that of the hatchery control group (0.2569), whereas the Gl of marked individuals was still lower than that of the hatchery control group (0.1080). In the third trimester, the Gl of marked individuals had increased to 0.0878, whereas the Gw (0.2815) remained almost the same.

After being released, the hatchery-produced S. wangchiachii began to move away from the release area. In October 2015, marked individuals were recaptured at all recapture sites upstream and downstream of the release sites (Table 2). Site 2 was not suitable as nursery ground for S. wangchiachii, but P2 (18.90%) nevertheless represented a high percent of total recaptures (Fig. 4). In the subsequent recaptures, catches at site 2 were very small because of unsuitable habitat (Table 2). In the three surveys, mean P3 (28.69 ± 1.03%) and P4 (31.62 ± 25.98%) were much higher than that of P5 (11.49 ± 7.80%), P6 (8.58 ± 4.38%) and P7 (4.17 ± 3.23%). Pi significantly decreased with distances from the release sites. Pi for the distant sites 6 and 7 in the latter two recaptures increased slightly compared to that in the first recapture, but the values were still much lower than those of sites 3–5 (Fig. 4). This implies that stocked fish were mainly distributed over a 10–15 km long stretch around the release sites.

Figure 4 Percent of all marked S. wangchiachii captured at each site in three recapture surveys in October 2015 (n = 164), January 2016 (n = 58), and April 2016 (n = 40).

(×, no recapture survey was conducted; *, the percent was 0%).

Discussion

Feasibility of ARS mass marking

Previous studies of marking different fish species demonstrated that ARS treatment produces an excellent mark quality and has no significant harmful effects on the fish (Baer & Rösch, 2008; Bashey, 2004; Caraguel et al., 2015; Liu et al., 2009). However, faced with sustained pressure to produce enough fish seed to achieve the annual goals of release programs, managers of many hatcheries continue to worry that mass marking using the ARS method will cause high mortality, and this concern has a negative impact on the use of marking to evaluate fish stock enhancement. In this study, marking juvenile S. wangchiachii (mean total length 23.52 ± 1.50 mm) by immersion in 70 mg L−1 ARS solution for 24 h did not cause death. In the following large-scale marking application, the mortality of five marked batches and three unmarked ones was negligible (≤0.50%), and no significant difference in mortality between marked and unmarked fish was detected. After rearing for three months in outdoor fishponds, no significant differences in total length, body mass, or condition factor between marked and unmarked groups were detected. Because juvenile S. wangchiachii experienced natural mortality, the extremely low mortalities that occurred during the immersion marking process might not have been due to ARS solution. In addition, immersion marking was carried out directly in the rearing tanks, which avoided manipulations of fish, reduced stress and costs.

The ARS mark that develops in the otolith remains highly readable for several years, whether fish are reared in the laboratory or in the field (Champigneulle & Cachera, 2003; Nagiec et al., 1995; Partridge et al., 2009; Poczyczyński et al., 2011). Because sunlight and turnover of skeletal calcium can cause fluorescent marks to fade, external fluorescent marks on scales and fin rays cannot be readily detected over time (Bashey, 2004; Elle, Koenig & Meyer, 2010). In contrast, otoliths are protected by the skull and previously deposited otolith materials are not resorbed (Campana & Neilson, 1985), which prolongs the lifetime of the mark. In the 2014 marking effort, S. wangchiachii marked with 70 mg L−1 ARS retained highly visible marks on otoliths after rearing for about 20 months in an indoor tank, and they did not present clear signs of significant fading. In the mass marking of 2015, fish marked with 30–50 mg L−1 ARS were transferred to outdoor fishponds and reared for about three months. Jinping Hatchery is located in the arid river valley region of the western Sichuan Province, where sunshine is very strong all year long (Yuan, Li & Lin, 2013). Nevertheless, ARS marks of fish sampled from each marking batch were highly visible. All of the ARS marks on the otoliths of recaptured fish were as clear as they had been at the time of release. However, because otoliths continuously grow and thicken, over time the mark can be covered by otolith materials, and marks can become faint and difficult to detect unless exposed by grinding and polishing the otoliths (Baer & Rösch, 2008; Sánchez-Lamadrid, 2001; Taylor, Fielder & Suthers, 2005). In this study, although it was not experimentally tested, the final retention time of the ARS mark in the otoliths of S. wangchiachii was long enough to monitor the released individuals to evaluate stocking effectiveness.

Effectiveness of stocking enhancement

After release, trimonthly recapture surveys confirmed that some of the stocked S. wangchiachii had survived. Assuming that the percent of marked (47.62%) and unmarked (52.38%) S. wangchiachii remained unchanged in the stocked cohort, the percent of catches that originated from stock enhancement were estimated to be 68.73% in October 2015, 55.35% in January 2016 and 64.11% in April 2016, respectively. This demonstrated that stocked S. wangchiachii constituted an important part of the young fish with a mean level of 62.73% recruitment of the 2015 cohort. In previous successful stock enhancements, such as those for vendace Coregonus albula (Poczyczyński et al., 2011), brown trout Salmo trutta (Caudron & Champigneulle, 2009), Japanese Spanish mackerel Scomberomorus niphonius (Obata et al., 2008), and red sea bream Pagrus major and Japanese flounder Paralichthys olivaceus (Kitada & Kishino, 2006), hatchery-produced fish contributed considerably to population recruitment, and stocking successes were often attributable to appropriate release sizes and environmental conditions at release sites. In this study, the relatively high percent of released S. wangchiachii might be explained by the fact that young fish for release were fully covered with scales, which would protect the skin against mechanical injury and bacteria and parasites (Yan et al., 2014). In addition, the negligible fishing pressure on young S. wangchiachii under age two and few predators, such as Percocypris pingi and Silurus asotus, might have had positive effects.

However, the comparative analysis of recapture data showed that total length, wet mass, and condition factor of recaptured S. wangchiachii differed significantly among different recapture sites (Table 3). The body sizes of fish recaptured at release sites were often smaller than those at other sites for both marked and unmarked fish (Table 2). This finding suggests that fish at release sites did not grow as well as fish at other sites. Pebbly nursery grounds in shallow waters are essential for Schizothorax fish at early life stages. In the Jinping area of the Yalong River, water flow sharply decreases (at a maximum percent of about 95%) due to the upstream dam of Jinping Dam II, which leads to marked physical habitat degradation (Wang et al., 2007). Excavation of sand in the river, which take place frequently at five sites along the 60 km long survey area, further destroyed the habitats. This reduction of essential habitats could have a significant negative effect on the river’s carrying capacity for Schizothorax fishes. It is likely that the released fish moved very slowly so that nine months after release most of them still were distributed over a 10–15 km long stretch near the release sites, although a few marked fish were caught about 50 km downstream three months after release. The relatively slow migration speed would maintain a high density of fish in the release area, which would result in both released and wild fish having to compete intensely with each other for resources.

Wild fish can be replaced with hatchery-reared fish when the latter are released in numbers that exceed the carrying capacity, but it is difficult to verify the extent to which they replace the wild ones (Kitada & Kishino, 2006). The surveys conducted in this study showed that, there are some spawning grounds for Schizothorax fishes in the Jinping area of the Yalong River, where some naturally born juveniles were caught in April 2014. This study showed that natural recruitment still accounted for a sizeable percent (approximate 40%) of S. wangchiachii recruitment of the 2015 cohort. Additionally, the non-significant difference in total length, wet mass and condition factor between the marked and unmarked fish indicated that ARS marking did not have significant harmful effects. This finding suggests that the stocked S. wangchiachii should grow as the naturally born fish in the Jinping area of the Yalong River. Releases at the scale used in this study might have not exceeded the carrying capacity of release sites or replaced wild fish, but repeated large annual releases might do so. To maintain the fish population at sustainable levels, new release sites around site 7 should be added (Fig. 1). Moreover, sand excavation in the river should be stopped immediately, and nursery habitats must be restored to expand the carrying capacity of the river.

Wild S. wangchiachii live in rapid flowing river water and consume adherent alga (Bacillariophyta) using the sharp outer horny sheath on their lower jaw to scrape it off the substrate. In contrast, the hatchery-reared fish are reared in still water ponds and fed on commercial diets. It is well known that environmental differences between hatchery-reared and wild fish can influence their behavior, especially foraging behavior and avoidance of predators, which may subsequently affect post-release success (Hervas et al., 2010; Le Vay et al., 2007; Johnsson, Brockmark & Näslund, 2014). On first release into the wild, hatchery-reared fish must not only avoid predators but also adapt to a new food supply (Blaxter, 2000). In this study, condition factors of the stocked S. wangchiachii recaptured three months after release were significantly lower than those of the hatchery control group as well as their own cohort at the time of release (Table 2). This suggests that these hatchery-produced fish might need to be acclimatized to the wild habitat, as their growth was negatively affected by the change of environmental conditions. Perhaps, hatchery-reared fish suffered a high level of short-term post-release mortality during the first trimester after release. However, it was difficult to precisely estimate this mortality due to lack of historical data on fisheries and prior investigation in the area. In the subsequent recaptures, the condition factor of stocked fish had returned to a level as good as that of the hatchery control group. In addition, the Gl and Gw of marked fish displayed a slowly increasing trend. Therefore, survival of hatchery-produced S. wangchiachii suggests that they gradually adapted to the wild habitat, and they exhibited favorable growth six months after release.

Conclusions

This study offers fishery administrators a cost-efficient method of mass marking juvenile S. wangchiachii with ARS. The marking process did not cause significant mortality or affect fish growth in this study. Release-recapture surveys indicated that the present stock enhancement might make a considerable contribution to the recruitment of young S. wangchiachii in the Jinping area of the Yalong River. Results of this study will be instrumental in promoting application of mass marking techniques and applying responsible approaches to the development of stock enhancement in China. However, much information about stock enhancement remains unknown, including the post-release mortalities of stocked fish, their contribution to the spawning population, and their genetic impact on the wild population. Therefore, in order to improve stocking strategy and better protect S. wangchiachii and other fish species in the Yalong River, long-term monitoring and further studies of the released fish should be conducted.

Supplemental Information

Table S1 Raw data 1

Click here for additional data file.

Table S2 Raw data 2

Click here for additional data file.

We thank Dr. Fangdong Zou for his technical assistance in examination of otoliths, and Dr. Eréndira Aceves Bueno at the University of California, Santa Barbara for his helpful revisions of English language.

Additional Information and Declarations

Competing Interests

Author Contributions

Animal Ethics

Data Availability

Xiaoshuai Liu, Longjun Deng and Weixiong Gan are employees of Yalong River Hydropower Development Company, Ltd., Chengdu, Sichuan Province, People’s Republic of China. The authors declare there are no competing interests.

Kun Yang conceived and designed the experiments, performed the experiments, analyzed the data, contributed reagents/materials/analysis tools, wrote the paper, prepared figures and/or tables, reviewed drafts of the paper.

Shu Li performed the experiments, analyzed the data, reviewed drafts of the paper.

Xiaoshuai Liu, Weixiong Gan and Longjun Deng performed the experiments, contributed reagents/materials/analysis tools, reviewed drafts of the paper.

Yezhong Tang wrote the paper, reviewed drafts of the paper.

Zhaobin Song conceived and designed the experiments, wrote the paper, reviewed drafts of the paper.

The following information was supplied relating to ethical approvals (i.e., approving body and any reference numbers):

Sichuan University Medical Ethics Committee

The following information was supplied regarding data availability:

The raw data has been provided as Supplemental Files.

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
