# Peer review of "Mass marking of juvenile Schizothorax wangchiachii (Fang) with alizarin red S and evaluation of stock enhancement in the Jinping area of the Yalong River"

_PeerJ, doi:10.7717/peerj.4142_

## Round 0.1 · original submission · Minor Revisions

The reviewers and I find your study a sound contribution to science and overall a well written and nicely presented study. We have a small number of suggestions to improve the clarity and contribution of your study. I have provided an annotated pdf with corrections to grammar and spelling indicated by highlights and inserted comments. I also checked the reviewer's comments on wording and have provided suggested changes to meet their concerns on the same pdf. In your rebuttal, you should reply to each comment by the reviewers and editor, indicating what changes you made. However, for the changes on the pdf, you need only mention suggestions that you disagreed with and did not change.

Editor's comments

Title: I suggest a simpler and more direct title: 'Using alizarin red S to evaluate stock enhancement of Schizothorax wangchiachii (Cyprinidae) '

Abstract
L29. Although Reviewer 1 would like more information about the meaning of the words 'slightly increase', I think it is acceptable for the abstract because details are available in the manuscript.

Methods
L95. For oxygen and pH, you provide a range. For temperature, it is not clear whether 16 ± 1.50 also refers to a range or SD. Clarify by either changing to a range (14.5 - 17.5) or indicating that the variation represents SD as you did elsewhere in the Methods (16 ± 1.50 oC, mean ± SD).
L129 Were the fish selected by a true randomization procedure? If not, you should use the term 'haphazardly' instead of randomly. Studies have shown that sweeping a net through a tank of fish results in selective rather than random capture. (The same issue arises at several other places in the Methods.)
L131. 'Body length' is not a standard term for fish measurement. Do you mean total length and standard length? Check the definitions and choose the terms that applies to the measurement you used. If you present data using only one length measure, you only need to mention that measure in your Methods.
L169. It is not clear what you mean by 'based on personal experience'. Do you mean 'based on size'?
L199. I agree with Reviewer 1 that you should indicate in the Methods which statistical tests you used for which analyses.

Discussion
L288. It is not clear what you mean by increased security.
L351. 'Live at the same degree' is unclear in this sentence. I think you mean that they grow at a similar rate. Are you also suggesting that they survive at a similar rate? This would be indicated by lack of change in the proportion of marked fish over time.

Tables
Table 2. I found Table 2 quite confusing at first, because the caption was incomplete, the headings of some columns were unclear and it appeared that two sample sizes were sometimes provided. I eventually realized that the sample sizes issue was due to the third digit moving to the next line. I have suggested changes to clarify the heading and body of the table. I only did this for July and October. Please apply the same corrections to the rest of the table. It is important not to allow numbers to be divided between two lines. However, if you need more space, you can put the standard deviations in parentheses below each mean without the need to include the +/- signs, as long as it is clear in the heading. It is not necessary to change the font size of the table as suggested by Reviewer 1. That can be addressed by the production staff.

Fig. 4. It might be helpful to state the number of recaptured fish in each survey in the caption as indicated in the pdf.

·

Basic reporting

The paper presents a detailed study on stock enhancement of S. wangchiachii and is written in a clear and concise manner. I would have liked to see a more precise definition of stock enhancement as well as a reference that shows such marking techniques are still clearly visible after several years in some species, e.g. in the otoliths of Acanthopagrus butcheri alizarin complexone was still visible after 15 years (Cottingham et al., 2015). I would also like to have more information on the species, such as maximum size and age and its importance as a targeted species.
The figures and tables are clear, however, I suggest smaller font on Table 2 would enable the table to be on one page instead of three. There are also some inconsistencies in spacing.

Experimental design

The experimental design was well thought out and executed and within the scope of the journal. The research question was well defined and fills an important knowledge gap. However, the description of the statistical analyses was not sufficient. This section needs to be expanded. Also, in the material and methods it needs to be reiterated that the means of the instantaneous growth and mass were calculated for different batches, otherwise it is not clear to the reader how statistical analyses can be undertaken.

Validity of the findings

The study demonstrates that restocking can be used as an effective measure to enhance the population of this species. The data was robust in that enough replicates were employed to undertake statistical comparisons.

It was concluded that this study offered a cost-efficient method for mass marking, but the costs are not discussed in the text.

Also, there is nothing in the introduction about total length, mass and condition factor and what these measurements represent, their meaning and why such measurements should differ between marked and unmarked fish.

Additional comments

I outline some general points below.
Use mm throughout instead of cm. No dash required between numbers and units.

Line 29. It is not clear what slightly increase means.
Line 43. otolith marking should be mentioned first as it is the most important in your paper. The same goes for fluorescent marking in line 46.
Line 137 Of the 600,000 marked fish, why were only 400,000 released.
Line 153/154 Consider rewording sentence.
Line 174 Three pairs, do you mean Sagitta, Asteriscus and Lapillus? If so, which ones did you mainly use.
Line 196/197 different font and double spacing
Line 229 The slight difference was statistically different, which one was less.
Line 234-242 What about condition factor between marked and unmarked fish.
Line 253 are the instantaneous rates unitless?
Line 323 What is the relevance of this?
Table 1 caption- treatments spelt wrong.
Figure 4 caption. Consider rewording

Reviewer 2 ·

Basic reporting

This ms fulfills the criteria given.
General comments: An interesting and most clear paper on an important problem, i.e. how to evaluate the outcome of massive restocking activities using young/small fish individuals.
My main concern, besides some details below, is that the paper is unnecessary long and complex. The main reason is that so much statistics is presented not only in the tables, but also in the running text. Please consider to omit most of those details in the text, but comment on them using a reasonable amount of details and “digits”.
I am not that familiar with this journal and it rules, so the following are most as questions and suggestions;
• As the same words and terms appear in both the title and as listed key-words, the key-words may be changed to increase the chance of being found in a search.
• Seems the references seem to come in the wrong order when several are given within brackets in the running text, i.e. the come in alphabetic order and not in time order that is the most? common way in scientific papers.

Experimental design

Seems OK with me. In a way there are even two control groups, one is the unmarked fish stocked in the river and the other one are the groups held in ponds, where I assume life and growth is more "comfortable".

Validity of the findings

OK, but as statistics is not my expertise I cannot judge that part in any details, but looks convincing to me. Seems the results are so obvious.

Additional comments

See above (1. Basic reporting)

Annotated reviews are not available for download in order to protect the identity of reviewers who chose to remain anonymous.

---

## Round 0.2 · accepted · Accept

The revisions have been carried out appropriately and the manuscript is now suitable for publication. However, a small number of suggested changes were not changed, possibly due to misunderstanding and a few grammatical errors remain to be corrected.

I have attached a manuscript with the problems highlighted and changes indicated in inserted comments (pp. 10, 11, 12, 15). On Table 2, in the first row, a missing decimal for a S.D. was corrected on the pdf but not on the table from the separate file. Also, in this table, I suggested that an extra row should be inserted under 'Hatchery Control Fish' in Oct, Jan and Apr. This should be labelled 'Recaptured Fish' because Sites 1 to 7 refer to recaptured fish. On the caption to Fig. 4, it is not necessary to have the % sign as well as Percent on the ordinate label. On L127, the authors should replace 'body length' with 'standard length' to make the measurement clear. The rebuttal indicated that the authors consider these measures the same, but they provided no reference and I have not seen the use of this term.